# Energy Optimization of the ‘Shore to Ship’ System—A Universal Power System for Ships at Berth in a Port

**DOI:** 10.3390/s20143815

**Published:** 2020-07-08

**Authors:** Sergey German-Galkin, Dariusz Tarnapowicz

**Affiliations:** Faculty of Mechatronics and Electrical Engineering, Maritime University of Szczecin, 70-500 Szczecin, Poland; s.german-galkin@am.szczecin.pl

**Keywords:** shore connection, “Shore to Ship” system, active and reactive power control, energy optimization, control of the active converter

## Abstract

One of the most effective methods of limiting air pollution emissions by ships at a berth in a port is the power connection of ships to the on-shore system. “Shore to Ship” (STS)—A universal system for the connection of the ship’s electrical power network with the on-shore network—ensures the adoption of the voltage and frequency of the on-shore network for the exploitation of various types of ships in the port. The realization of such a system is possible due to the use of semiconductor technologies during the construction of mechatronic systems (i.e., systems that ensure the maintenance of electricity parameters). The STS system ensures energy efficiency for high-power ship systems through the use of an active semiconductor converter. This article presents an analysis of steady state electromagnetic and energy processes, allowing the determination of the active and reactive power and losses in the STS system. The presented analytical research enables the development of a control algorithm that optimizes the system energy efficiency. In the article, the control methods allowing the optimization of the energy characteristics of the system are considered and investigated. On the basis of theoretical studies, a model was developed in the Matlab-Simulink environment, which allowed us to study steady and transient processes in the STS system in order to reduce losses in power lines and semiconductor converters.

## 1. Introduction

When mooring ships in a port, a source of electricity is required (i.e., for the operation of all systems). For this purpose, autonomous diesel generator (D-G) sets are commonly used. Depending on the power demand, from one to several D-G units can operate in a ship mooring at a quay. Each of these units can burn up to several tons of fuel every single day. D-G generating sets emit toxic compounds into the atmosphere. When analyzing various sources of air pollution in the port, it was stated that seagoing vessels are the main source [1,2,3,4,5,6].

Low-emission ports are an essential condition for global sustainable development. One of the most effective methods to reduce the emission of air pollution by ships is to use a “Shore to Ship” (STS) system; i.e., a universal system for the connection of the ship’s electricity network with the on-shore network [1,6,7,8,9].

When the ship’s power supply is provided from the onshore power grid, the ship’s generating sets, which are the main source of air pollution and noise in the port, are turned off. When the generator sets are turned off, ship crews and port workers will benefit due to noise reduction (for example: the operation of D-G sets of 1 MW generates noise level of 140 dB) [1]. Apart from the environmental aspect related to the exclusion of autonomous ship generating sets, the crew has the possibility to carry out their repair and inspection. Another important aspect of shore-side deliveries is the reduction in electricity costs, as shore-side electricity is cheaper than energy from ships’ generating sets [1].

The versatility of the STS system for the connection of various types of vessels to the on-shore power network requires global standardization of the system.

The main problem in the technical implementation of the STS system is connected with matching the frequency and voltage level of the ship’s power network and the on-shore network [1,7,8,9]. In 2006, the European Union (EU) recommended the construction and development of systems for the collection of electricity from the on-shore network by ships moored in ports [10]. In accordance with the EU recommendations, member states (during the building of systems) should follow the technical solutions presented in the annex to these recommendations (chapter: Technical requirements—typical configuration). The chapter concerning technical requirements defines the main elements of the STS system. Figure 1 shows the configuration of the STS system based on the annex to these recommendations.

The presented configuration of the STS system does not introduce technical details for the construction of the system but only presents a very general solution. In 2012, the IEC (International Electrotechnical Commission), ISO (International Organization for Standardization) and IEEE (Institute of Electrical and Electronics Engineers) developed a global standard that enabled the standardization of the design and construction of STS systems [11,12]. This standardization was intended for ships with a power demand above 1 MW and it was based on high-voltage (HV) installations. In 2014, the standardization of STS systems for ships with a power demand below 1 MW (based on low-voltage (LV) installations) was developed [12,13]. There may be many topologies of the STS system that meet the described standards [13,14]. Figure 2 presents an exemplary topology of the system, which is in accordance with the IEC/ISO/IEEE standardization.

In addition to the main feature of universality, the STS system should ensure uninterrupted switching from the ship’s electricity network to the on-shore network and vice versa. The system is switched with the use of synchronization systems [15,16,17,18]. The switching of systems (national power grid to ship power grid) should be flexible; i.e., without dynamic load changes. When switching from the ship network to the on-shore network, power should be gently transferred from the ship’s generating sets; therefore, the STS system must be able to convert power in both directions.

The article considers and examines the methods for the control of an active semiconductor converter that is able to optimize the energy performance of the system. A similar solution can be used for on-shore power networks.

## 2. Functional Diagram of the ‘Shore to Ship’ (STS) Mechatronic System

The main elements of the analyzed STS system are two active converters connected with each other with the use of a direct current (DC) circuit (with a capacitive filter). The general functionality of the system is presented in Figure 3.

The elements of the system (Figure 3) include the following:AFC—Active frequency converter;AC_1, AC_2—Active converters;ACCS 1, ACCS 2—Control system of active converters;${\bar{E}}_{1},{\bar{U}}_{1},{\bar{I}}_{1}$—Electromotive force, voltage and current vectors in the AC_1;${\bar{E}}_{2},{\bar{U}}_{2},{\bar{I}}_{2}$—Electromotive force, voltage and current vectors in the AC_2;$r_{1},\;X_{1}=\omega_{1}L_{1}$—Resistance and reactance in the alternating current circuit of the AC_1 converter$r_{2},\;X_{2}=\omega_{2}L_{2}$—Resistance and reactance of the ship load (or synchronous generator of the ship load)$U_{dc},C$—Voltage and capacity in the direct current circuit.


The operation of AC_1 is synchronized with the network. The angular (fundamental $\omega_{1}=2\pi f_{1}$) frequency of this network is determined by the automatic phase frequency regulation system (PLL). The operation of AC_2 is synchronized with the generating set on the ship. For example, in systems with synchronous generators, a rotor position sensor (SPS) is used for this purpose. In such a case, the angular frequency is dependent on the mechanical speed of the shaft ($\omega_{2}=p\omega_{m}$). AC_2 can also be connected to a passive load. In such a case, the electromotive force (E2) is replaced by the voltage drop on the load.

If the STS mechatronic system transfers active power from the on-shore network to the ship’s network, AC_1 acts as an active rectifier, and AC_2 acts as an inverter. When the STS mechatronic system transfers active power from the ship’s network to the on-shore network (when switching the ship’s electrical network to the on-shore network), AC_1 acts as an inverter, and AC_2 is an active rectifier. This two-way energy conversion capability (B2B—Back to back) is the main advantage of this system; an additional advantage is its ability to reduce losses to provide a sinusoidal form of current in the electrical network.

In the common DC circuit, only active power (*P_dc_*) is transmitted. In AC circuits, active power (*P*_1_, *P*_2_) is transmitted, while reactive power (*Q*_1_, *Q*_2_) is exchanged in the AC circuits of AC_1 and AC_2 converters. The active powers *P*_1_, *P*_2_ are rigidly connected with each other (equal in the case of the negligence of losses in AFC), and the reactive power depends on the method of controlling the AC_1 and AC_2 converters. These powers are not dependent in any way and can be controlled independently of each other (decoupled control).

In [19,20,21], methods of controlling active converters in an electric drive with a synchronous machine ensuring maintenance and zero reactive power were considered and analyzed.

In this article, the optimal control methods proposed in [19,20,21] were extended with the control of the AC_1 active converter connected to the on-shore network in the STS system.

Analytical and modeling tests were conducted on the basis of the analysis of electromagnetic processes in quasi-steady modes, described in the classical works of scientists in the field of electrical engineering, electromechanics and automatic control theory [22,23,24,25,26,27,28,29].

## 3. Analysis of the STS System with Independent Control of AC_1 Active Converter

The mathematical description of electromagnetic processes in the STS system is realized with the use of the spatial vector method [23,24] and method of fundamental components [25,26].

In relation to the mains, the active converter AC_1 (Figure 3) can be represented by electromotive force. Its fundamental component is equal to [27]
(1)$$\begin{array}{l} {\bar{E}}_{1}=\frac{m}{2}U_{dc}e^{-j\varphi_{m}}=E_{1y}-jE_{1x},\\ E_{1x}=\frac{m}{2}U_{dc}\sin\varphi_{m},\\ E_{1y}=\frac{m}{2}U_{dc}\cos\varphi_{m}. \end{array}$$
where:

*m*—The modulation factor;

$\varphi_{m}$—The shift angle between the network voltage vector ${\bar{U}}_{1}$, and electromotive force vector of AC_1 ${\bar{E}}_{1}$ (modulation angle);

$E_{1x},E_{1y}$—The value of electromotive force in relation to the x and y axis.

The angle $\varphi_{m}$ depends on the control signal and the load (current in the circuit). If the modulation factor *m* and angle $\varphi_{m}$ are considered as control signals, they can be set in the control system to a certain extent and used to examine the electromagnetic and energy characteristics of the system. This method of controlling the inverter is called independent control.

Testing the STS system with an independent control method enables the determination of the limits for status variables and properties.

The equation for electromagnetic processes in a determined operating mode based on Kirchhoff’s second law for the fundamental component of the STS system (Figure 3) can be presented in the following way:(2)$${\bar{U}}_{1}={\bar{E}}_{1}+r_{1}{\bar{I}}_{1}+jX_{1}{\bar{I}}_{1}$$

In order to explain the energy properties of the considered system, vector analysis was introduced. The STS vector graph in the synchronously rotating coordinate system (x, y), synchronized with the network, is presented in Figure 4. In Figure 4a, the vector graph corresponds to the case in which AC_1 operates in the mode of an active rectifier. For the case in which AC_1 operates in an inverter mode, the vector diagram of the system is shown in Figure 4b.

In addition to the *x*, *y* coordinates connected with the network voltage ${\bar{U}}_{1}$, so that ${\bar{U}}_{1}=U_{y},U_{x}=0$, in Figure 4, the rotating axes *d*, *q* (connected with the electromotive force of the active converter CA_1) were marked in order to ensure that ${\bar{E}}_{1}=E_{q},E_{d}=0$. In the steady state, both coordinate systems are fixed to each other. The AC_1 converter is controlled by the ACCS_1 control system in *x*, *y* axes.

In relation to the mains, the STS system consumes active power when $\varphi_{m}<0^{0}$ (i.e., AC_1 acts as the active rectifier) or transfers active energy to the network $\varphi_{m}>0^{0}$ (i.e., AC_1 acts as the inverter). The system with independent control is described by a system of non-linear equations. This system has problems connected with the stability of operation in a certain range of changes of control signals. Converter current control should be used for the reliable and stable operation of the STS system.

## 4. Analysis of the STS System with Current Control

In the active converter connected to the network via a parametric choke (*r*_1_, *X*_1_), current control is usually realized when the current in CA_1 is generated with the use of negative feedback from the transmitter [28,29]. In such a case, the set (control) signals are network currents (*I_x_*, *I_y_*) in the x and y axes. Electromagnetic and energy characteristics in the STS system with current control are calculated with the use of Equations (3) and (4) prepared on the basis of the geometrical relations of the vector graph presented in Figure 4.
(3)$$\begin{array}{l} \varphi_{m}=-\mathrm{arc}tg\frac{X_{1}I_{y}-r_{1}I_{x}}{U_{1}-(X_{1}I_{x}+r_{1}I_{y})},\\ E_{1}=\sqrt{{\left(U_{1}-r_{1}I_{y}-X_{1}I_{x}\right)}^{2}+{\left(X_{1}I_{y}-r_{1}I_{x}\right)}^{2}},\\ E_{x}=E_{1}\sin\varphi_{m},\\ E_{y}=E_{1}\cos\varphi_{m}, \end{array}$$
(4)$$\begin{array}{l} P_{1(1)}=1.5U_{1}I_{y},\\ Q_{1(1)}=1.5U_{1}I_{x},\\ \Delta P_{1(1)}=1.5r_{1}(I_{x}^{2}+I_{y}^{2}),\\ P_{AC(1)}=1.5(E_{x}I_{x}+E_{y}I_{y}),\\ Q_{AC(1)}=1.5(E_{y}I_{x}-E_{x}I_{y}),\\ \Delta P_{AC(1)}=1.5r_{AC}(I_{x}^{2}+I_{y}^{2}). \end{array}$$
where:

*P*_1(1)_, *Q*_1(1)_—the active and reactive power in the power network;

*P_AC_*_(1)_, *Q_AC_*_(1)_—the active and reactive power in the AC_1 active converter;

Δ*P*_1(1)_, Δ*P_AC_*_(1)_—the power losses in the network and in the AC_1 converter;

*r*_1_—the resistance of the whole STS system;

*r_AC_*—the equivalent resistance of the AC_1 converter.

The results of analytical tests presenting the energy characteristics for the STS system calculated on the basis of Equations (3) and (4) are presented in Figure 5, Figure 6 and Figure 7 as dependences of active power *P*_1(1)_ and reactive power *Q*_1(1)_ in the load network (Figure 5), as dependences of active power *P_AC_*_(1)_ and reactive power *Q_AC_*_(1)_ in the active converter AC_1 (Figure 6) and as power losses in the whole STS system and the converter (Figure 7). The calculated energy characteristics calculated and constructed during the change of currents are *I*_x_ and *I*_y_.

When evaluating the energy properties of the system with current control, the following conclusions can be drawn:Active power in the network only depends (linearly) on the current *I*_y_, while reactive power in the network only depends (linearly) on the current *I*_x_. Active and reactive power can have positive and negative values (Figure 5).Active and reactive power in the active converter depends (non-linearly) on the currents *I*_x_ and *I*_y_. Powers can have positive and negative values (Figure 6).Power losses in the STS system and in the AC_1 active converter (parabolically) depend on currents *I*_y_ and *I*_x_.

## 5. Analysis of the STS System with Optimized Current Control

The analysis of the energy performance of the STS system with current control AC_1 [19,20] proves that, with a certain dependency between control signals (*I*_x_, *I*_y_), it is possible to maintain reactive power in the supply network at a level of zero, regardless of the value and direction of active power flow transmitted by the STS system. This operating mode is determined as the optimized mode. It should be stressed that the optimized operating mode of the network does not coincide with the optimized mode of the AC_1 active converter. This difference results from the presence of a choke in the common circuit with parameters *r*_1_, *X*_1_.

For the analysis of the STS system with optimized power consumption from the network, Figure 8 presents vector charts with a phase shift between current and voltage in the network of 0° (AC_1 is an active rectifier) or 180° (AC_1 is a network inverter).

The control of the active converter (AC_1) is realized with the use of a PLL (phase-locked loop) in x and y axes. In this case, the condition of optimality is met when ${\bar{I}}_{1}=I_{1}=I_{y},I_{x}=0$. Electromagnetic and energy characteristics are calculated on the basis of Equations (5) and (6) received from the geometrical dependencies of the vector diagram (Figure 8). They have the following form:(5)$$\begin{array}{l} \varphi_{m}=\mathrm{arc}tg\frac{X_{1}I_{1}}{U_{1}-r_{1}I_{1}},\\ I_{d}=I_{1}\sin(-\varphi_{m}),\\ I_{q}=I_{1}\cos(-\varphi_{m}),\\ E_{1}=\sqrt{{\left(U_{1}-r_{1}I_{1}\right)}^{2}+{\left(X_{1}I_{1}\right)}^{2},} \end{array}$$
(6)$$\begin{array}{l} P_{1(1)}=1.5U_{1}I_{1},\\ Q_{1(1)}=0,\\ \Delta P_{1(1)}=r_{1}I_{1}^{2}.\\ P_{AC(1)}=1.5E_{1}I_{q},\\ Q_{AC(1)}=1.5E_{1}I_{d},\\ \Delta P_{AC(1)}=1.5r_{AC}(I_{d}^{2}+I_{q}^{2}). \end{array}$$

The energy characteristics of the STS system for the optimization of the power consumption from the network, calculated with the use of Equations (5) and (6), are presented in Figure 9.

During the optimization of power consumption from the network, the reactive power in the AC_1 converter is negative, both in the mode of an active rectifier and in the mode of a network inverter. This means that the network for the AC_1 converter is an active–capacitive load. In this case, the electromotive force (*E*_1_) is greater than the voltage *U*_1_ (Figure 8).

Current control in the AC_1 converter takes place within the limits determined by the ratio of the *U*_1_ network voltage and electromotive force at the AC terminals of the *E*_1_ converter. In the time interval in which the instantaneous valve $e_{1}$ exceeds the instantaneous valve $u_{1}$, the relay current regulator (in the control system) goes into the state of saturation and loses the ability to switch the converter’s transistors. During this time, there is no switching to the converter’s branches. This enables the relative time during which the converter’s transistors do not switch to be calculated, which can be presented (approximately) in the following way:(7)$$\omega t≃\pi -2\arcsin\frac{U_{1}}{E_{1}}≃\pi -2\arcsin\frac{U_{1}}{\sqrt{{\left(U_{1}-r_{1}I_{1}\right)}^{2}+{\left(X_{1}I_{1}\right)}^{2}}}$$

The presence of such an interval leads to a reduction in losses of $\Delta P_{AC}$ in semiconductor elements of the converter; this enables the calculation of the relative coefficient for the loss reduction:(8)$$\lambda ≃1-\frac{2}{\pi }\arcsin\frac{U_{1}}{\sqrt{{\left(U_{1}-r_{1}I_{1}\right)}^{2}+{\left(X_{1}I_{1}\right)}^{2}}}$$

The dependency of the relative loss reduction coefficient in relation to the current for the STS-optimized system is presented in Figure 10. In abscissae, the current is shown in relative units (pu) defined as the ratio of real current *I*_1_ to short-circuit current $I_{sc}=\frac{U_{1}}{z_{1}}$:(9)$$I_{pu}=\frac{I_{1}}{I_{sc}}=\frac{z_{1}I_{1}}{U_{1}}$$

Power losses in the AC_1 converter (taking into account the relative loss reduction coefficient, *λ*) are determined in accordance with the following formula:(10)$$\Delta P_{AC(\lambda ,1)}=(1-\lambda )\Delta P_{AC(1)}$$
where:

$\Delta P_{AC(1)}$—Power losses in the AC_1 converter determined on the basis of Equation (6).

The analysis of the STS energy characteristics can be conducted taking into account the losses in the AC_1 converter to the fundamental component, $\Delta P_{AC(1)}$, and the reduction in switching losses *λ* in the converter, $\Delta P_{AC(\lambda ,1)}$.

For the final conclusion regarding losses in the AC_1 converter, it is necessary to calculate not only the fundamental component (taking into account the reduction in switching losses, *λ*), but also the power losses at all other fundamental components that occur during the switching of transistors to the carrier frequency. These losses will be called switching losses.

When calculating switching losses in the active converter, a distinction should be made between currents in the converter phase and currents in semiconductor elements of the same converter phase. Currents in phases are responsible for power transfer; these currents are sinusoidal, because the transmission of power takes place at the fundamental frequency (modulation frequency). Currents in semiconductor elements are responsible for energy exchange. The exchange is carried out with the carrier frequency. A significant distortion of currents has an impact on switching losses, which can be taken into account with the use of the total harmonic distortion (THD) according to the following formula:(11)$$\Delta P_{AC(\lambda ,n)}=\left[1+{\left(THD\right)}^{2}\right]\Delta P_{AC(1)}$$
where:

$\Delta P_{AC(\lambda ,n)}$—Switching losses in the AC_1 converter, taking into account all fundamental components, including the loss reduction coefficient, *λ*.

In the current control, the switching frequency (carrier frequency) in the AC_1 converter depends on the choke time constant ($\tau_{1}=\frac{L_{1}}{r_{1}}$), the width of the hysteresis loop of the relay current regulator in the control system and the instantaneous value of the current *I*_1_. The switching frequency value for the systems (medium and high power) is in the range of 1–10 kHz.

The calculation of all energy indicators cannot be realized in an analytical way. This is still implemented with the use of imitation (virtual) models in the Matlab-Simulink environment.

## 6. STS System Studies

The block diagram of the STS system is shown in Figure 11.

The main task of the STS system is electric energy transfer from the onshore grid to the ship’s grid. In this mode of operation of the STS system the AC_1 converter works as an active rectifier. The optimization block (OB) ensures the energy optimization of the system. As a result of the transformation of set currents $I_{x}^{*}$, $I_{y}^{*}$ to the three-phase a, b, c coordinate system the preset currents $I_{a,b,c}^{*}$ were obtained, which in the tracking system (hysteresis) will provide a constant voltage value *U_dc_* in the DC circuit. An analogous control algorithm is realized in the ACCS_2 system. Switching between on shore grid and ship grid systems in transition states ensures controlled (soft) transfer of active power between both systems.

The model of the examined system is presented in Figure 12.

The control of transmission in both directions of active power *P*_1_ and reactive power *Q*_1_ in the network is realized by current *I_x_* and *I_y_* (Figure 12; Ix-set point, Iy-set point). The study compared a system with and without optimized power transmission control. The AC_1 converter is built as a secondary voltage stabilization circuit on the capacitor in the DC circuit (Figure 11) [20,27]. In order to minimize the calculations in the simulation model (Figure 12), a DC voltage source was used in the DC circuit, which allowed the possibility of control of active power flow in both directions to be checked, with a possibility to check the idea of energy optimization of the STS system. The use of a DC voltage source in a simulation model provided the control concept shown in the block diagram of Figure 11.

The results of tests for the optimized STS system are indicated in Figure 13, Figure 14, Figure 15 and Figure 16.

The system is tested in two steady state conditions:The state of power transmission from the mains to the ship’s network. In this case, active power is transferred from the on-shore network to the ship’s network. The AC_1 converter operates in the mode corresponding to the active rectifier.The state of power transmission from the ship’s network to the network. In this case, active power is transferred from the ship’s network to the on-shore network. The AC_1 converter operates in the mode corresponding to the network inverter.

The system is tested in a transient status (with changes in the direction of active power’s transfer).

The energy processes in the system (*P*_1(1)_, *Q*_1(1)_, *P_AC_*_1(1)_, *Q_AC_*_1(1)_) are presented in Figure 13. In the determined operating modes of the system, the reactive power in the network *Q*_1(1)_ is zero. A change in reactive power is observed during the transition of AC_1 from the active rectifier mode to the network inverter mode (Figure 13). The reactive power in the converter (in steady states) is always negative and changes only in the transient state.

The power losses in the optimized STS system in steady and transient states in the case of changes in the direction of active power transmission are shown in Figure 14.

The electromagnetic processes in the *U*_1_, *E*_1_, *I*_1_ system in steady and transient states when switching AC_1 from the operating mode corresponding to the active rectifier to the operating mode corresponding to the inverter are presented in Figure 15. In steady states, electromagnetic variables are sinusoidal.

Figure 16 presents the waveforms of voltage and current for the semiconductor elements in one branch of the AC_1 converter in the optimized STS system for two operating modes corresponding to different directions of active power in the STS system.

The results of tests for the non-optimized STS system are presented in Figure 17, Figure 18, Figure 19 and Figure 20.

The system is tested in the state of active power transfer from the on-shore network to the ship’s network. The system is tested in a steady state and in a transient state during the change of reactive power.

Energy processes in the system (*P*_1(1)_, *Q*_1(1)_, *P_AC_*_1(1)_, *Q_AC_*_1(1)_) when switching the AC_1 from the operating mode corresponding to positive (inductive) reactive power to the operating mode corresponding to negative (capacitive) reactive power are presented in Figure 17.

Power losses in the STS system in the discussed operating modes are presented in Figure 18. Compared with Figure 15, where *Q*_1(1)_ = 0, power losses in steady states increased by more than 10%.

The electromagnetic processes in the system *(U*_1_, *E*_1_, *I*_1_) when switching AC_1 from the operating mode corresponding to positive (inductive) reactive power (*Q*_1(1)_ > 0) to the operating mode corresponding to negative (capacitive) reactive power (*Q*_1(1)_ < 0) are presented in Figure 19.

Voltages and currents in semiconductor elements (*U_VT_*_1_, *I_VT_*_1_, *U_VT_*_2_, *I_VT_*_2_) for AC_1′s branch are presented in Figure 20. Figure 20 shows that, in the case of *Q*_1(1)_ > 0, there is no gap in the switching processes of semiconductor elements. In the case in which *Q*_1(1)_ < 0, the switching processes of semiconductor elements do not occur in a time interval lasting approximately 0.4 of the period’s duration.

The current spectrum of one semiconductor element in the optimized STS system (where operation of AC_1 corresponds to the active rectifier) is presented in Figure 21a.

The current spectrum of one semiconductor element in the STS system (at *Q*_1(1)_ > 0 corresponding to the active rectifier) is presented in Figure 21b.

Table 1 presents a comparison of the STS system with and without energy optimization in terms of power losses in the transmission line (Δ*P*_1(1)_), power loss (fundamental component) in the AC_1 converter (Δ*P_AC_*_(1)_) and switching losses in AC_1 (Δ*P*_1_*_AC_*_(*λ,n*)_).

The formulas (4), (11) and the results of the tests presented in Figure 14, Figure 18 and Figure 21 were used in the calculations. The calculations were made for the resistance of the transmission lines *r*_1_ = 6 × 10^−4^ Ohm and constant active power *P*_1(1)_ = 1.2 MW. The results are presented for the STS system without optimization *Q*_1(1)_ > 0 (*I_y_* = 2500 A, *I_x_* = 1000 A) and with optimization *Q*_1(1)_ = 0, (*I_y_* = 2500 A, *I_x_* = 0 A).

It should be noted that for STS systems of higher power (e.g., for STS systems supporting passenger ships, the transferred power is from a dozen to several dozen MW) and with a lower power factor, the value of the proposed system’s energy optimization will be more significant.

## 7. Conclusions

The basic problem in the design and implementation of the STS network is connected with the increase in energy efficiency. Currently, the energy efficiency, normally understood as the ratio of active output power to active input power (*P*_2_/*P*_1_), is relatively large, and it is difficult to expect significant improvement. However, enterprises (companies and production plants) incur costs that are not only connected with active power consumption but also with reactive power consumption.

As mentioned above, it is difficult to minimize costs related to active power consumption. The issue of reactive power compensation is still being developed. Apart from reactive power compensators (constructed from capacitors—passive compensators (PC)), it is also possible to build active compensators (AC), which use active converters (energy electronic converters).

Energy optimization in the STS system (which reduces the reactive power in the supply network to zero) reduces the loss in the entire system for the fundamental components and decreases switching losses on the carrier frequency in the semiconductor elements of the active converter by approximately 15–30%.

In the active converter (when switching semiconductor elements), there is an increase in switching losses proportional to THD. These losses should be taken into account during the calculation of energy indicators.

In addition to the optimization of the above quantities, active converters are also able to change the direction of power flow. The discussed conclusions were confirmed by the results of analytical and model tests; our contributions in this paper are, therefore, as follows:A method of analysis of steady state electromagnetic and energetic processes in the “port power grid–active converter” circuit has been developed.An analytical method has been developed to determine the active and reactive power in the port power supply network and active converter, as well as to determine power losses in this circuit.The optimal control of the active converter was synthesized when the reactive power in the port power supply network was equal to zero.For the developed optimal control algorithm, the power in the port’s power supply network and in the active converter was calculated.The developed method of calculating switching losses was based on the spectral analysis of currents in semiconductor elements of the active converter.A simulation (virtual) model of the “port power network–active converter” system has been developed.

The comparison of the simulation results with theoretical results allows us to recommend the developed methods of analysis and calculation for the design of an STS in a wide power range, as well as for use in medium-voltage STS systems. Transmission of reactive energy through power grids has many negative effects, such as voltage drops, heating of cables, and reduction of capacity of infrastructure used for energy transmission. The importance of the proposed increase in the energy efficiency of the STS system (the study examined an STS system with a power of about 1 MW) is growing for systems with higher power due to the economic effects.

It should also be emphasized that the proposed strategy to control the converter with energy optimization has reduced transistor power loss during switching by reducing the number of switches over the period. This increase in the converter’s efficiency results in the improvement of the efficiency of the whole STS system, which is particularly important for high-power systems.

## Figures and Tables

**Figure 1 sensors-20-03815-f001:**
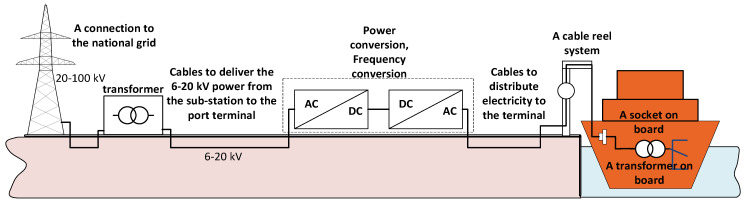
Configuration of the “Shore to Ship” (STS) system in accordance with the European Union (EU) recommendations [10].

**Figure 2 sensors-20-03815-f002:**
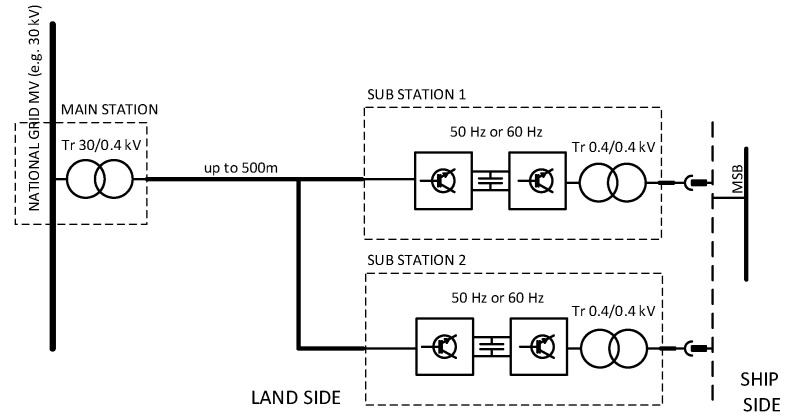
Selected topology of the STS system in accordance with the International Electrotechnical Commission (IEC)/International Organization for Standardization (ISO)/Institute of Electrical and Electronics Engineers (IEEE) standard.

**Figure 3 sensors-20-03815-f003:**
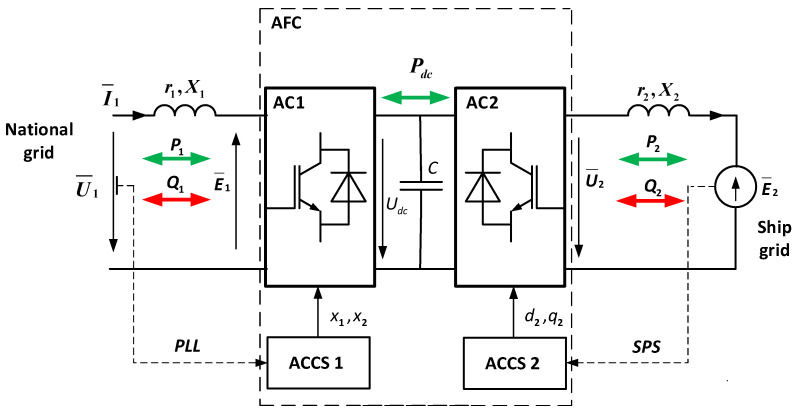
Functional diagram of the STS mechatronic system. AFC: Active frequency converter; AC: Active converter; ACCS: Control system of active converters.

**Figure 4 sensors-20-03815-f004:**
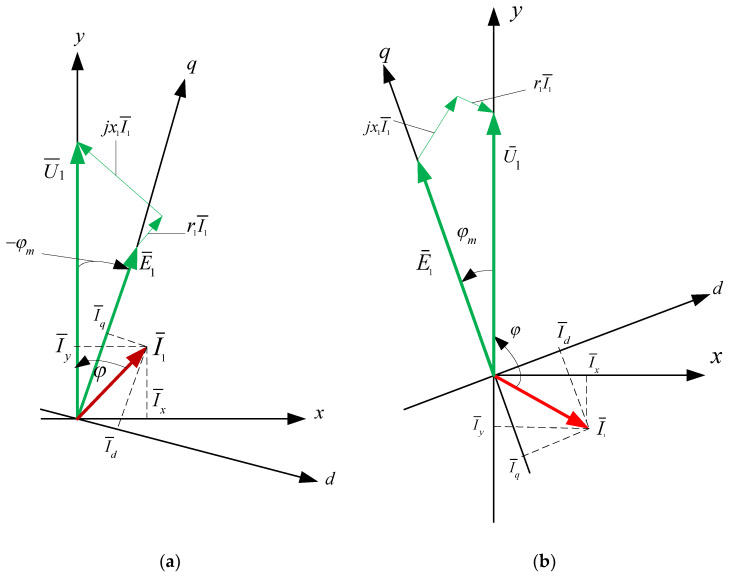
Vector diagram of the STS system, when AC_1 operates in a mode of an active rectifier (**a**) and inverter (**b**).

**Figure 5 sensors-20-03815-f005:**
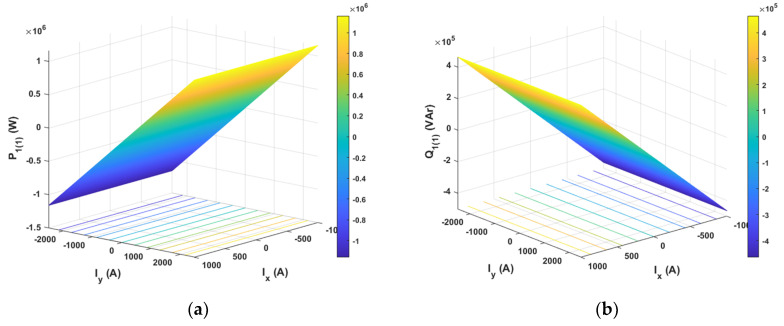
Active (**a**) and reactive (**b**) power of the fundamental component in the STS power supply network with current control.

**Figure 6 sensors-20-03815-f006:**
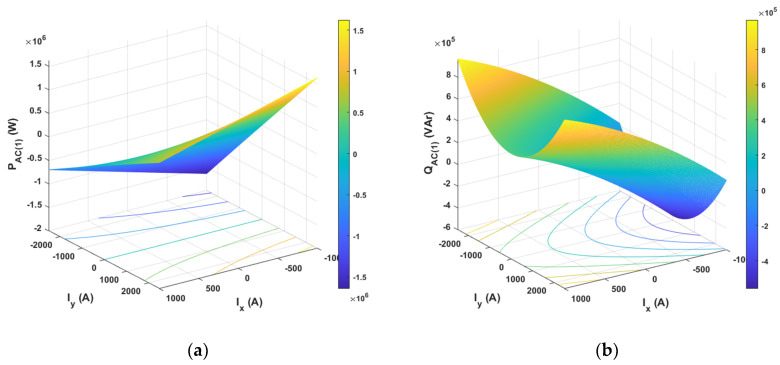
Active (**a**) and reactive (**b**) power of the fundamental component in the AC_1 converter with current control.

**Figure 7 sensors-20-03815-f007:**
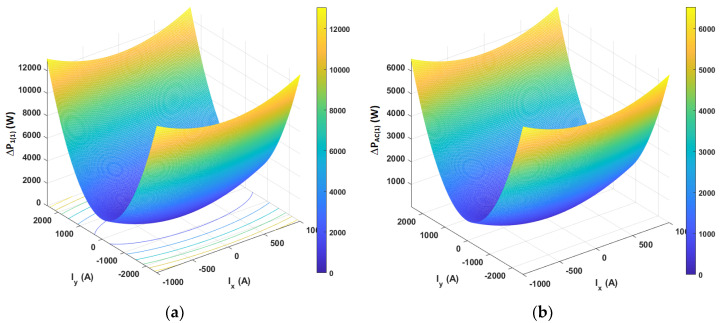
Power losses in the fundamental component (STS system) in the whole STS system (**a**) and in the AC_1 converter (**b**).

**Figure 8 sensors-20-03815-f008:**
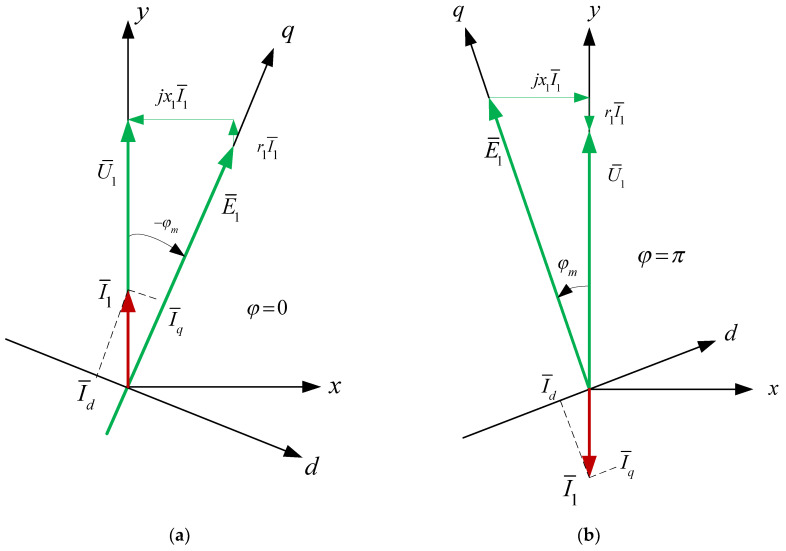
Vector charts of the optimized STS system when AC_1 operates (**a**) in the mode of an active rectifier and (**b**) in the mode of a network inverter.

**Figure 9 sensors-20-03815-f009:**
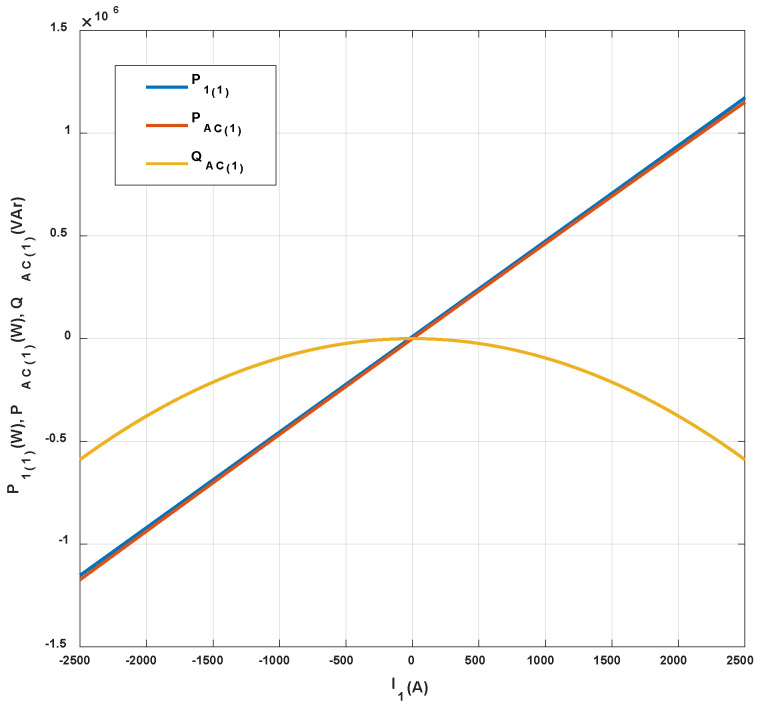
Energy characteristics of the optimized STS system.

**Figure 10 sensors-20-03815-f010:**
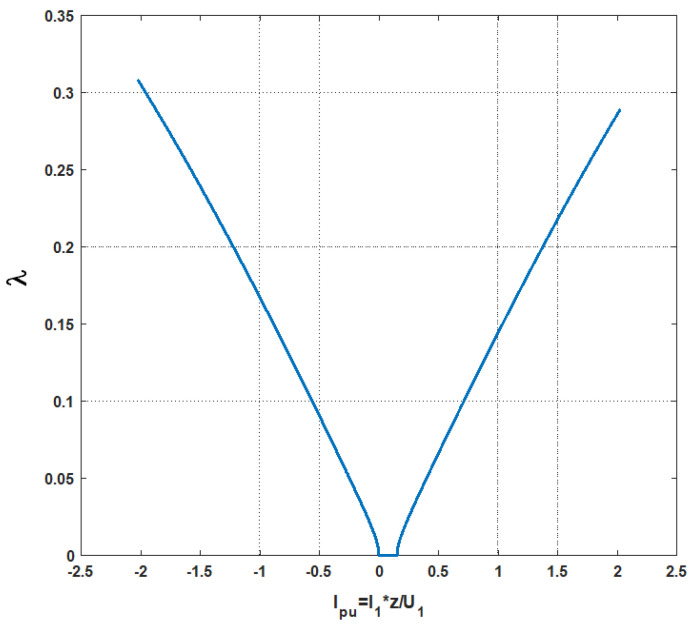
Dependence of the relative loss reduction coefficient *λ* in the function of the current of the active converter.

**Figure 11 sensors-20-03815-f011:**
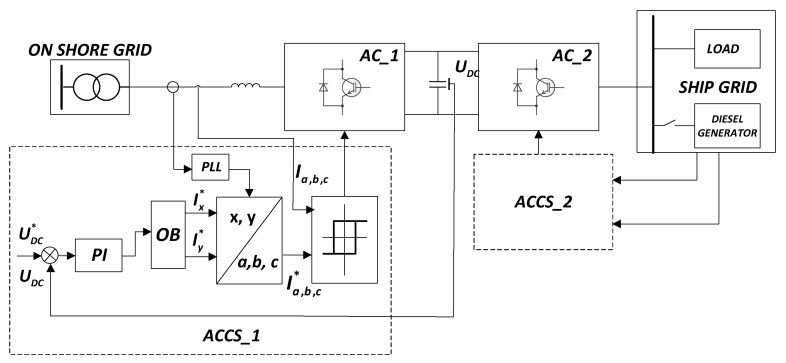
Block diagram of the STS system.

**Figure 12 sensors-20-03815-f012:**
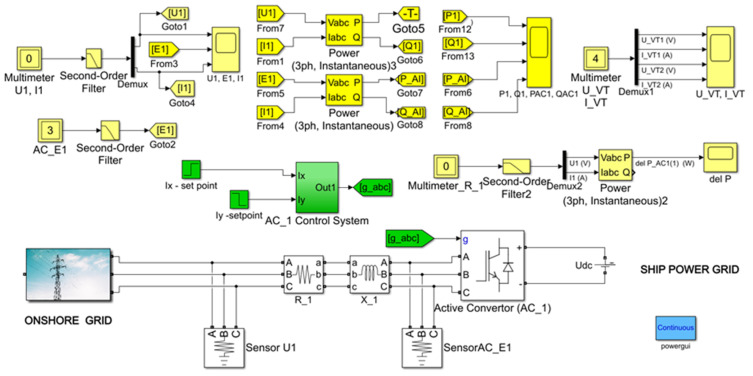
Simulation model of STS system.

**Figure 13 sensors-20-03815-f013:**
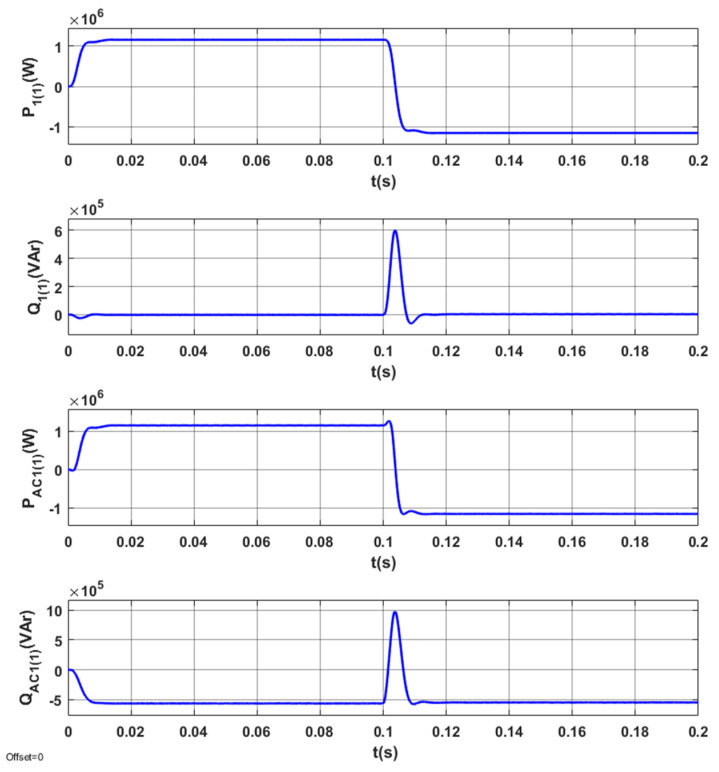
Energy processes *P*_1(1)_, *Q*_1(1)_, *P_AC_*_1(1)_, *Q_AC_*_1(1)_ in the optimized STS system, when switching AC_1 from the operating mode corresponding to the active rectifier to the operating mode with the network inverter.

**Figure 14 sensors-20-03815-f014:**
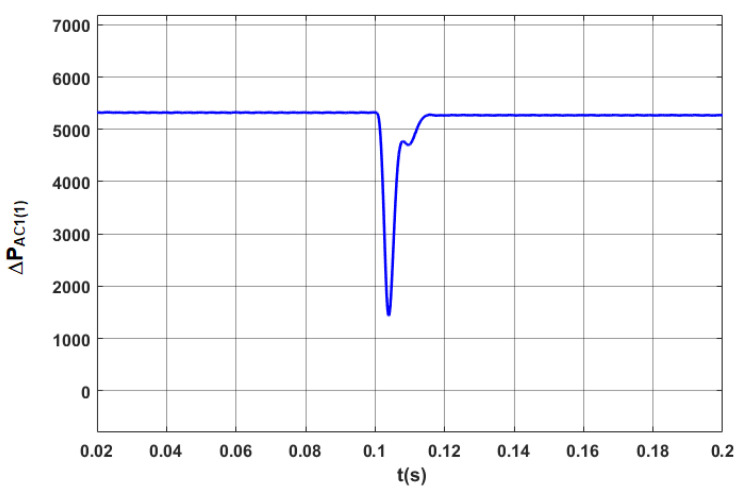
Power losses in the AC_1 converter ($\Delta P_{AC(1)}$) in the optimized STS system for the fundamental component, when switching the AC_1 from the operating mode corresponding to the active rectifier to the operating mode with the network inverter.

**Figure 15 sensors-20-03815-f015:**
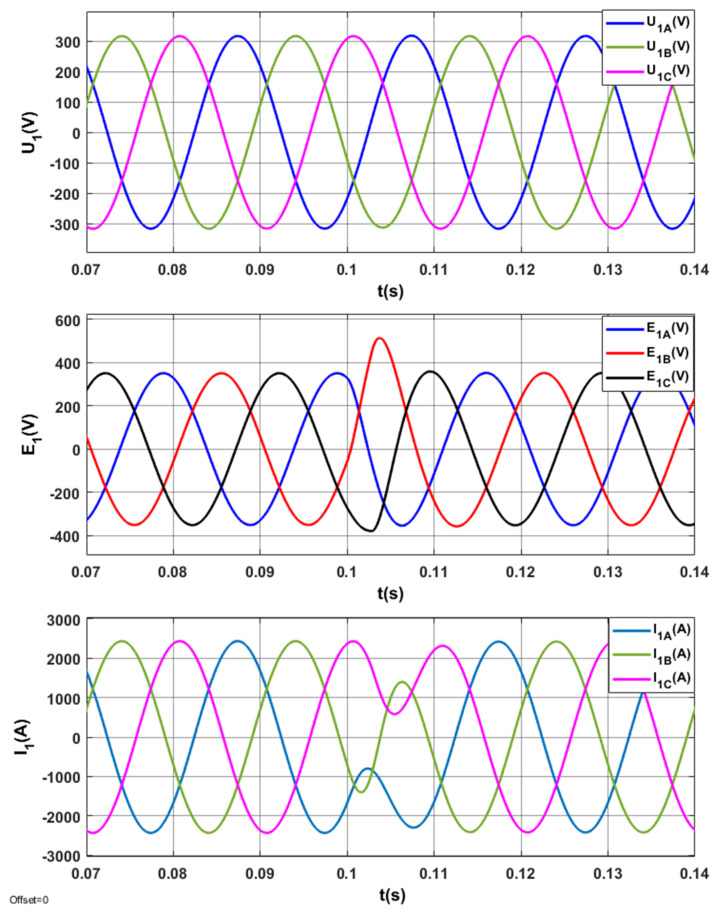
Electromagnetic processes in the optimized STS system, when switching the AC_1 from the operating mode corresponding to the active rectifier to the operating mode with the network inverter.

**Figure 16 sensors-20-03815-f016:**
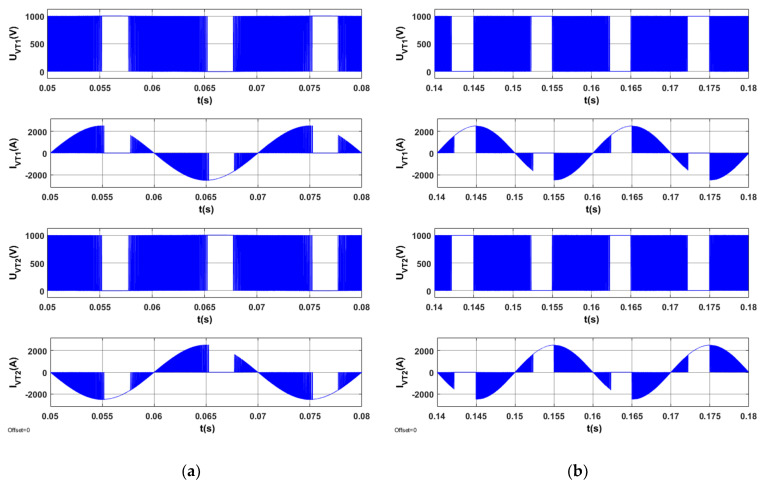
Voltages and currents on semiconductor elements of AC_1′s branch in the optimized STS system in the operating mode of an active rectifier (**a**) and in the operating mode of a network inverter (**b**).

**Figure 17 sensors-20-03815-f017:**
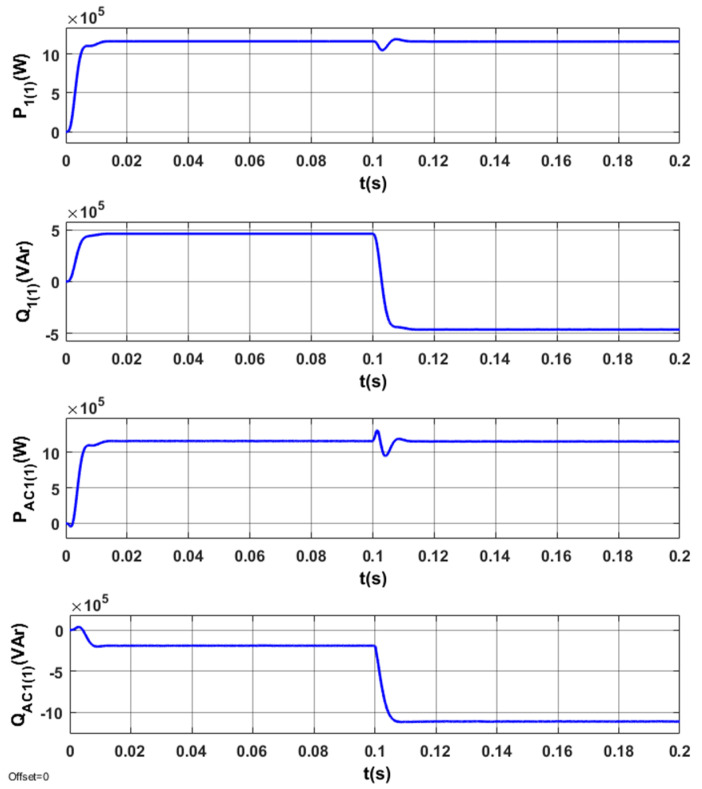
Energy processes *P*_1(1)_, *Q*_1(1)_, *P_AC_*_1(1)_, *Q_AC_*_1(1)_ in the STS system, taking into account the operating mode of the AC_1 corresponding to the active rectifier (*I_y_* = 2500 A) and *I_x_* changing from 1000 A to −1000 A.

**Figure 18 sensors-20-03815-f018:**
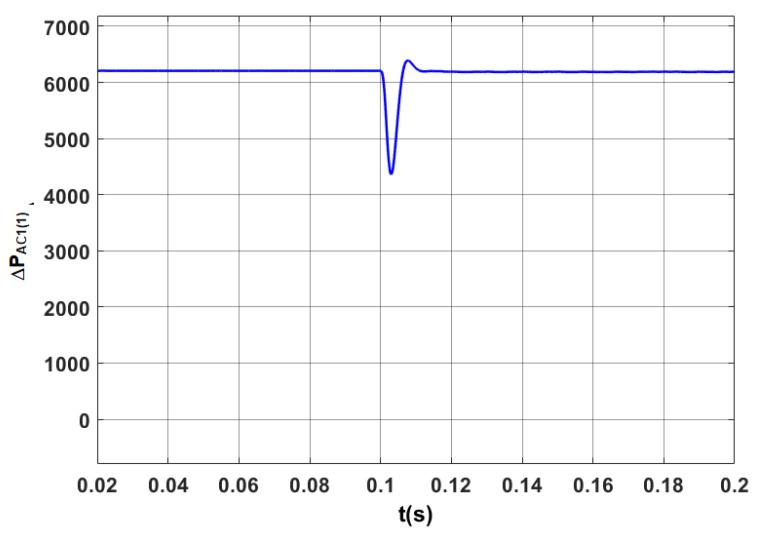
Power losses for the fundamental component in the STS system, taking into account the operating mode of the AC_1 corresponding to the active rectifier (*I_y_* = 2500 A) and *I_x_* changing from 1000 A to −1000 A.

**Figure 19 sensors-20-03815-f019:**
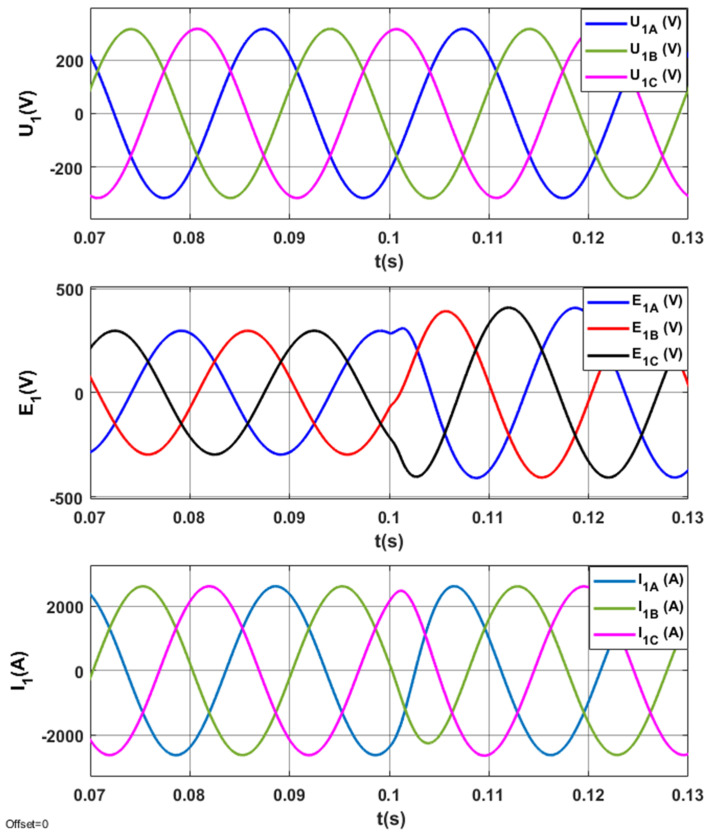
Electromagnetic processes (*U*_1_, *E*_1_, *I*_1_) in the STS system corresponding to the active rectifier (*I_y_* = 2500 A) and *I_x_* changing from 1000 A to −1000 A.

**Figure 20 sensors-20-03815-f020:**
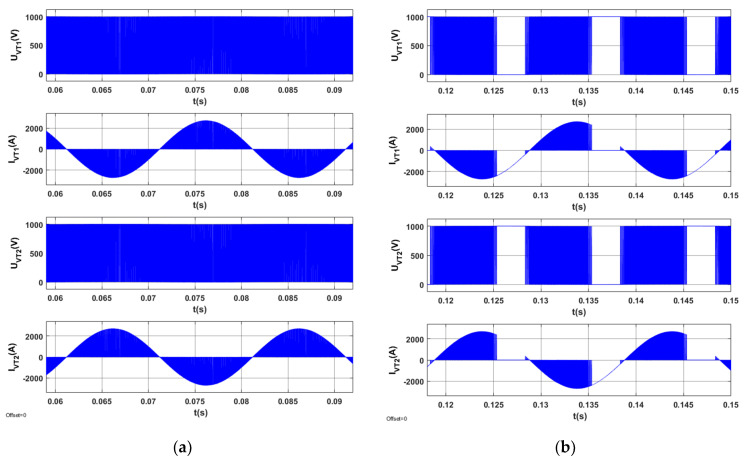
Voltages and currents on semiconductor elements of AC_1′s branch, which operates in the active rectifier mode in the STS system, (**a**) *Iy* = 2500 A, *Ix* = 1000 A, (**b**) *Iy* = −2500 A, *Ix* = −1000 A.

**Figure 21 sensors-20-03815-f021:**
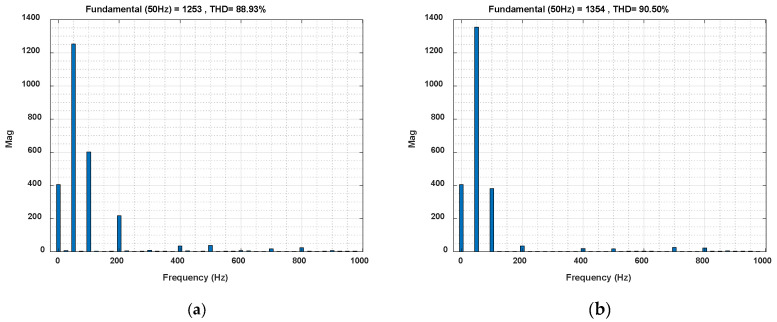
Current spectrum in the semiconductor element of the AC_1 converter operating in the active rectifier mode with the optimized STS system (**a**) and *I_y_ =* 2500 A, *I_x_ =* 1000 A (**b**) for two of the 10 cycles of the selected current signal *I_VT_*_1_.

**Table 1 sensors-20-03815-t001:** Comparison of power losses in STS system with and without energy optimization.

Power Losses	Power Losses in the STS System without Optimization [W]	Power Losses in the STS System with Optimization [W]	Reduction of Losses through the Use of Optimization [%]
Δ*P*_1(1)_	6525	5625	13.8
Δ*P_AC_*_(1)_	6210	5320	14.3
Δ*P*_1_*_AC_*_(*λ,n*)_	11296	9527	15.7

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
