# Peer review of "Energy Optimization of the ‘Shore to Ship’ System—A Universal Power System for Ships at Berth in a Port"

_sensors, 2020, doi:10.3390/s20143815_

Round 1

Reviewer 1 Report

Dear authors,

This paper presents an analysis of the “Shore to Ship” system with the optimized current control. The paper is well written in the beginning with a relatively interesting introduction, but at the end of the paper, there is less description, with poor comparative analysis with a baseline system. The systems will come more and more in the future in order to reduce emissions at harbor from ships. In Norway they install many of such systems today. Such system are already used in airplanes when they come to airports. Please describe better the operation at harbor – when reading the txt it sounds the engines will still be running – with the risk of air-polution but the idea is that the ships can turn off the engines or what ?

The first thing was the size and fonts of the figures which are very different. Further, figures 12, 13, 14, 16, 17, 18 do not have signed axes and dimensions. Please improve them. Also, in figures 14, 18 sinusoidal waveforms with three colors, did not show what voltages and currents it is? Moreover, in the text authors describe Figure 19, but in the paper Figure 19 do not exist.

It would be interesting to know the presented modified “Shore to Ship” system can be implemented for which power for ships above 1 MW or below 1 MW. In addition, the conclusions are very crumpled; the results of the study were not deeply analyzed and well explained – so put more effort into doing that. expressed and proved by the authors. Please highlight that better.

Author Response

the review response in the additional file

Reviewer 2 Report

All comments are given in the attached file

Author Response

(The authors gave the same response as above.)

Round 2

Reviewer 1 Report

Thanks a lot for the update of the paper. In my opinion, the authors took not in to account all recommendations and suggestions to improve paper about 'shore to Ship" system and did not answer all my questions.

In addition, the simulated system – by showing a Simulink file is not easy to read. Please make your own block diagram. The analysis seems to be on a simplified system with a battery – why not a full system which will be much more interesting and better for the paper.

Author Response

response for reviewer included in the attached file

Round 3

Reviewer 1 Report

Nevertheless, the paper should be improved in the case of a comparative analysis with the base system and more precise confirm the optimization of energy consumption of the "Shore-Ship" system.

Author Response

The reviewer's answer in the appendix
